# Implantation Failure in Endometriosis Patients: Etiopathogenesis

**DOI:** 10.3390/jcm11185366

**Published:** 2022-09-13

**Authors:** Astrid Boucher, Géraldine Brichant, Virginie Gridelet, Michelle Nisolle, Stéphanie Ravet, Marie Timmermans, Laurie Henry

**Affiliations:** 1Center for Reproductive Medicine, University of Liege, Boulevard du 12ème de Ligne 1, 4000 Liege, Belgium; 2Obstetrics and Gynecology Department, University of Liege, Boulevard du 12ème de Ligne 1, 4000 Liege, Belgium

**Keywords:** endometriosis, infertility, implantation disorders, adenomyosis

## Abstract

Embryo implantation requires adequate dialogue between a good quality embryo and a receptive endometrium. This implantation is still considered as the black box of reproductive medicine. Endometriosis is a highly prevalent chronic inflammatory disease, concerning about 10% of women of reproductive age and is one of the major causes of female infertility. The mechanisms involved in endometriosis-related infertility, an event not yet completely understood, are multifactorial and include anatomical changes, reduction in ovarian reserve, endocrine abnormalities, genetic profile, immunity markers, inflammatory mediators, or altered endometrial receptivity. In this article, we will focus on the impact of endometriosis on embryo quality and on endometrial receptivity. Results: Poor oocyte and embryo quality seem to promote a lower pregnancy rate, more than the endometrium itself in women with endometriosis. Other studies report the contrary. In addition, hormonal imbalance observed in the endometrium could also alter the embryo implantation. Conclusions: Controversial results in the literature add difficulties to the understanding of the mechanisms that lead to embryo implantation disorders. Furthermore, either oocyte/embryo impairment, altered endometrium, or both may cause impaired implantation. New prospective, randomized, and controlled studies are necessary to determine the origin of the defects that make conception more difficult in the case of endometriosis and adenomyosis.

## 1. Introduction

Endometriosis, defined as the presence of endometrial tissue outside the uterine cavity [1], is a chronic inflammatory disease that affects reproductive-aged women. It presents in various forms, from just a few implants on the pelvic peritoneum to extensive adhesions and organ infiltration, even outside the pelvis (i.e., pleural, diaphragmatic, umbilical). Three different phenotypes, often associated with each other, are described and are as follows: superficial peritoneal lesions, ovarian endometriomas and deep infiltrating endometriosis [2].

The clinical presentation is variable, from asymptomatic to pelvic pain, dysmenorrhea, dyspareunia, or dyschesia, seriously affecting the patient’s quality of life. Most patients with endometriosis struggle with infertility. The link between these two medical entities, although clinically recognized, is complex and implies multiple intricate pathways.

The exact prevalence of endometriosis in the population is difficult to determine. It ranges from 10% to 15% [3]. In subfertile women, the prevalence seems to be considerably higher, ranging from 20% to 50%, but with significant variation over time and with the age of patients [4,5]. In women with endometriosis, 30–50% were reported to experience infertility [6]. The accumulated scientific and clinical evidence suggests that reproduction in women remains a rather selective and somewhat inefficient process. The normal monthly fecundity rate (couple’s probability of conceiving in one month) is only around 15–20% in healthy couples, much less than that observed in other mammalians species. Endometriotic women have a monthly fecundity rate of 2–10% [7]. Even minimal endometriosis may be associated with marked subfertility [8]. In a large cohort study on women of reproductive age, the risk of infertility approximately increased 2-fold in women below 35 years with endometriosis, compared to women without endometriosis [9]. In assisted reproductive technologies (ART), patients with endometriosis have statistically significant lower pregnancy and implantation rates per cycle and per transfer (versus a control group consisting of women with tubal infertility, undergoing in vitro fertilization (IVF)) [10]. Arici et al. showed low implantation rates in patients with endometriosis, especially in patients with minimal-mild endometriosis [8]. In contrast, a study observed that although women with severe endometriosis showed lower ovarian responses and lower fertilization rates compared to women with tubal factors, embryo cleavage, implantation and clinical pregnancy rates are similar in these patients, suggesting that once the oocyte is fertilized, the pregnancy likelihood is comparable between groups [11]. Why some women with endometriosis display infertility and others do not is still currently unknown. Understanding the involved mechanisms would be of great help in improving patients’ fertility.

This review aims to highlight the possible mechanisms underlying embryo implantation in patients affected by endometriosis with possible associated adenomyosis, which are as follows: oocyte alterations, hormonal imbalance, endometrial receptivity and progesterone resistance.

## 2. Materials and Methods

The PubMed database (National Library of Medicine, https://pub-med.ncbi.nlm.nih.gov/, accessed on 12 April 2022) was searched for articles indexed until April 2022, with the use of the following medical subject heading (MeSH) search terms: “endometriosis” and “embryo implantation”, with restriction to the human species and to the English language. Title and abstracts were examined and full articles that met the selection criteria were retrieved. One hundred and twenty-four articles were excluded, as they were irrelevant and would not contribute to the article. Twenty-one publications were selected [7,12,13,14,15,16,17,18,19,20,21,22,23,24,25,26,27,28,29,30,31]. A manual screening completed the research.

## 3. Oocyte Alterations

Oocyte quality, and consequent embryo quality, are key factors in the chances of implantation and development of an evolutive pregnancy.

The hypothesis of altered oocyte quality in patients with endometriosis is frequently raised. A specific and delicate balance between inflammatory factors and sex hormones (i.e., estrogen and progesterone) is required for the follicular development and production of a fully competent oocyte, which can reach the mature MII stage and be fertilized, leading to embryo implantation and then, pregnancy. During follicle growth, an antrum is formed and filled with follicular fluid (FF). This fluid and its components are key factors to the success of natural fertilization. FF is composed of many substances, such as hormones, cytokines, and immune cells (including natural killer cells, lymphocytes, and macrophages), enzymes, anticoagulants, electrolytes, reactive oxygen species, lipids, cholesterol, and antioxidants. Although their physiological role in the normal follicle is not yet completely determined, all those factors may influence the oocyte, and its competence acquisition [32]. Indeed, the oocyte undergoes growth and maturation in the antral follicle, accumulating mRNA, mitochondria, lipids and proteins that are necessary for the segmentation phase of the embryo. If these reserves are not properly built, the beginning of embryonic development cannot proceed. 

In several studies, endometriosis was characterized by the presence of the local pelvic inflammatory process [7,33]. Whether the dysfunction of the immune system initiates or is the consequence of the condition is yet to be ascertained. The immune system of women with endometriosis, associated or not with infertility, is different from healthy and fertile controls [7]. The follicular microenvironment may be altered due to the dysregulation of molecular mechanisms [7]. Moreover, the granulosa cells surrounding the oocyte have higher apoptotic incidence, more alterations of the cell cycle and superior incidence of oxidative stress than in women whose infertility is due to other pathologies [25]. Endometriotic patients with oocyte donation from healthy donors had the same implantation and pregnancy rate as women without endometriosis, suggesting that oocytes from affected patients may compromise fertility [10]. Moreover, in women included in an oocyte donation program whose donated oocytes came from women with Stage III and IV endometriosis, implantation was significantly reduced [10,16]. Indeed, Simón and al. retrospectively examined 178 embryo transfers (ET) into 141 recipients divided into the following 3 groups: patients with premature failure (*n* = 54), low responders to conventional ovarian stimulation for IVF (*n* = 77), and women with endometriosis (*n* = 10). There was no difference among the groups in the rates of pregnancy per patient, per cycle or implantation [10]. To further investigate the influence of endometriosis on the endometrium or the oocyte, the same team analyzed the outcome of 701 embryos transferred in 179 cycles, according to the origin of the donated oocytes (fertile, polycystic ovaries, idiopathic infertility, tubal infertility, male infertility, and endometriosis). There was a trend towards reduced pregnancy rates in women receiving oocytes from endometriotic donors (27.3%), as compared to the remaining groups (44.0–60.3%). When the implantation rates were compared, there was a significantly lower result when the oocytes derived from endometriotic patients (7.0% versus 11.2–23.6%) [10], arguing that embryos derived from the oocytes of women with the disease are of lower quality, and contribute to a lower implantation rate [16]. Blank et al. confirmed the impaired oocyte and embryo characteristics in women with endometriosis, compared with couples with male subfertility in their matched cohort study in 2020 [22]. Borges et al. observed that a poorer quality of the oocytes and embryos could be an explanation for the lower implantation rates observed in patients with endometriosis [27]. A review on how endometriosis affects oocyte quality and embryo development was published in 2021. It highlights that oxidative stress, as well as the immune system dysregulation, constitute the two major pathophysiological mechanisms affecting oocyte and embryo competence in patients with endometriosis [34]. Nevertheless, there seems to be no increase in the risk of chromosomal abnormality, since the aneuploidy rate is equivalent between patients with and without endometriosis of the same age in the IVF population [35]. 

In contrast, by controlling the embryo quality in using euploid frozen ET (FET) cycles, Bishop et al. found no difference in pregnancy outcomes in patients with endometriosis (compared with patients undergoing treatment for male factor infertility and no infertile patients) [36].

In conclusion, several studies indicate that poor oocyte and embryo quality promotes lower pregnancy rates, more than the endometrium itself in women with endometriosis. However, the results remain heterogeneous in studies concerning patients with different types of endometriosis (ovarian or not), operated or not. Future studies are necessary for further comprehension of the endometriosis pathophysiology.

## 4. Endometrial Microenvironment and Endometrial Receptivity

Uterine receptivity represents the ability of the endometrium to allow normal embryo implantation [13]. Abnormal uterine receptivity leads to a range of reproductive problems, from complete implantation failure (infertility) to severely deficient implantation (miscarriage). Whether the primary cause of implantation failure lies in the mother or the embryo is a longstanding and still unresolved question [14].

Histologically, uterine receptivity is established during the mid-luteal phase, also called the window of implantation (WOI), with the phenomenon of decidualization that occurs each cycle independently of the presence of the embryo [20,29]. Decidualization is a complex and intense remodeling of the endometrium, involving endocrine, paracrine, and autocrine factors. This leads to the establishment of an endometrial environment capable of enhancing embryo implantation, protecting the embryo against the maternal immune rejection, and promoting interstitial and endovascular trophoblast invasion of the uterine junctional zone [20]. This process can be disturbed by endometriosis, involving the dysregulation of relevant signaling pathways and molecules in endometrial stromal cells, differential endometrial gene expression, alterations in cell physiology, vascular abnormalities, etc. [20]. A prospective comparative study showed the result of 240 cycles, placing sibling oocytes from the same donor into menopausal women with or without endometriosis. They showed reduced implantation, clinical pregnancy, ongoing pregnancy, and live birth rates (23.81% vs. 31.48%, 45.00% vs. 58.33%, 37.50% vs. 50.83%, 35.00% vs. 50.83%, respectively) for women with endometriosis, compared with the control group [37]. Early studies on endometrial proteins that participate in embryo attachment and invasion reported a decrease in the expression of key proteins associated with endometriosis [12,38,39].

Inflammation is known to alter endometrial receptivity and has been associated specifically with endometriosis [38]. Several immunologic disturbances that involve the number and/or activity of selected immune cells, particularly uterine natural killer cells (uNK), are present in the endometrium. Numerous other endometrial molecules have been suggested to play a role, including cell adhesion molecules (CAMs), cytokines, growth factors, steroid hormones, and others. Some have been hinted to characterize the complex signature of optimal receptivity, but none have been widely incorporated into clinical practice, given their poor ability to predict pregnancy [23].

CAMs are thought to play a substantial role in endometrial implantation. Indeed, during implantation, the embryo adheres to the endometrium via CAMs present on the blastocyst trophectoderm and endometrial epithelial cells. The CAM family is composed of the following four members: integrins, selectins, cadherins, and immunoglobulins. These are most likely regulated by hormones, cytokines, and growth factors. Integrins, principal adhesion molecules, are glycoproteins that serve as receptors for extracellular matrix (ECM) ligands. They are composed of two subunits (α and β) and are considered as primary mediators of adhesion on the apical surface of trophoblast cells. They adhere to various ECM components or to another surface receptor type (such as other CAMs). Some integrins are closely connected to cell surface proteins, such as matrix metalloproteinases. Several are critical biomarkers for endometrial receptivity, being highly specific to the initiation of the WOI on postovulatory day 5 to 6. Abnormal expression of cellular adhesion molecules in the eutopic endometrium is proposed to impair implantation in endometriosis patients. For example, estrogen has been shown to inhibit the expression of the αvβ3 integrin in the endometrial epithelium [38]. However, αvβ3 integrin, expressed only during cycle days 20 to 24 [12], initiates the cell–cell interactions to attach the trophoblast to the uterine epithelium. Lack of αvβ3 expression is associated with delayed histological development or out of phase endometrium, potentially disturbing fertility [14,25]. This integrin is decreased in women with infertility and endometriosis [40]. 

The pro-implantation cytokine, leukemia inhibitory factor (LIF) is essential for normal implantation [38]. Studies showed its reduced expression in women with endometriosis. Other key molecules that are required for normal endometrial receptivity, such as HOXA 10, are reduced in endometriosis [38].

uNK represent 40% of the total leukocyte population during the proliferative phase in the normal human endometrium, but increases to 60–70% by the mid-secretory phase, at the time of embryo implantation [15,30]. Once activated, uNK produce angiogenic factors, such as the vascular endothelial growth factor (VEGF), that promote spiral arteries remodeling and cytokine secretion, which create a pro-invasive environment in the endometrium, leading to the implantation of the trophoblast [14,15,30]. In women with endometriosis, findings suggest that endometrial leukocyte populations are altered, with conflicting results [17]. When these cells are dysregulated in number and/or cell function (cytotoxicity, receptor expression, cytokine secretion, or gene expression), the ability of the uterus to correctly regenerate the endometrium and/or eliminate shed endometrial cells is likely to be dysregulated as well. This could be critical for embryo tolerance and lead to implantation failure. A study with 61 patients revealed that patients with an increased number of uNK are at a greater risk for infertility disorders [15,30]. However, Freitag et al. showed no increased risk of an elevated number of uNK in women with endometriosis, compared to the control group. Their results instead suggest that patients with endometriosis are 1.3 times more likely to have chronic endometritis [15]. 

In ART, there remains a debate as to the effects of endometriosis on the outcomes of IVF treatment. Bishop et al. found no difference in pregnancy outcomes in patients with endometriosis (compared with patients undergoing treatment for male factor infertility and no infertile patients), when using FET [36]. In contrast, a retrospective cohort study demonstrated that endometriosis reduces the chance of a live birth in women undergoing ART (fresh ET), an effect that is more pronounced with increasing disease severity [41]. Given the possible impairment in endometrial receptivity due to the supra-physiologic hormonal levels observed during conventional stimulation, could the “freeze-all strategy” restore optimal receptivity in endometriosis-affected women and increase the pregnancy rate [42]? A large retrospective study showed that the freeze-all strategy led to a significantly higher implantation rate, clinical pregnancy rate and live birth rate compared to fresh ET transfer (advanced endometriosis) [43]. A retrospective matched cohort study, including 135 endometriosis-affected women in the fresh ET and deferred ET group, respectively, reported significantly higher cumulative clinical pregnancy rates (43.0% vs. 29.6%) and cumulative ongoing pregnancy rates (34.8% vs. 17.8%) in the deferred ET, group compared to the fresh ET group [42]. 

In conclusion, strong evidence supports the concept that endometrial defects exist in women with endometriosis. However, since there are conflicting observations between the studies, further research is needed to fully clarify how the presence of endometriotic lesions affects decidualization [20,21,29]. In ART, the “freeze-all” strategy seems to be an attractive option in endometriosis-affected women to increase success rates. However, further studies are necessary to confirm more firmly these results.

## 5. Hormonal Imbalance

The adhesion and invasion of the blastocyst require the precise coordination of embryonic and endometrial physiology and modulation of maternal immune tolerance, since 50% of the embryo is paternal material, and therefore immunologically foreign material. Endocrine and immune systems are among the most essential regulators of endometrial physiology. 

It is well established that endometriotic implants are dependent of estrogen for their maintenance and growth. Although peripheral levels of estrogen are not different between women with or without endometriosis, the amount of local estrogen to which endometriotic tissue is exposed exceeds systemic levels. This is due to the overexpression of P450 aromatase, an enzyme that converts androstenedione to estrone, an inactive steroid, leading to increased concentrations of estradiol, in ectopic or eutopic endometrium [38].

Another abnormal change in women with endometriosis is the increased expression of nitric oxide synthase (NOS) in the endometrium. It produces nitric oxide (NO), a vasodilator synthesized from L-arginine. NO is a free radical with cytotoxic effects, promoting cellular apoptosis in the endometrium. Increased NO levels may directly cause infertility due to their cytotoxic effects. Up-regulation of NOS may occur through increased inflammatory cytokines levels in patients with endometriosis, and high local estrogen levels. Local upregulation of NO production induces activation of the cyclooxygenase type 2 (COX-2) enzyme [25]. COX-2 catalyzes the conversion of arachidonic acid to prostaglandin E2 (PGE2), leading to its overproduction. PGE2 was found to be the most potent inducer of aromatase activity in endometriotic cells. Moreover, estrogen has been shown to stimulate PGE2 formation by stimulating COX-2 enzyme expression. PGE2, produced in excess in the eutopic endometrium and endometriotic tissues, contributes to dysmenorrhea and pelvic pain and, along with cytokines, to infertility. In a positive feedback loop, PGE2 coordinately stimulates the expression of all steroidogenic genes, favoring continuous local productions of estrogen and prostaglandins. Finally, endometriotic cells lack 17β-hydroxysteroid dehydrogenase type 2, leading to an increased local concentration of estradiol. Indeed, 17β-hydroxysteroid dehydrogenase type 1 catalyzes estrone into the biologically active form of estrogen, estradiol, to exert a full estrogenic effect, whereas 17β-hydroxysteroid dehydrogenase type 2 has the opposite effect, metabolizing estradiol to estrone [7].

In conclusion, decreased expression of 17β-hydroxysteroid dehydrogenase type 2, aberrant expression of COX-2 and overactivated P450 aromatase increase estradiol and PGE2 in ectopic implants and endometrium in women with endometriosis. This hormonal imbalance observed in the endometrium of endometriotic patients, with a state of estrogen dominance, promotes lesion growth, impaired follicular steroidogenesis, alteration of the proliferative to secretory phase transition, and therefore interferes with normal embryo implantation [7,14].

## 6. Progesterone Resistance

Besides leukocytes, pro-inflammatory cytokines and CAMs, and aberrant production of sex hormones, an abnormal expression of the sex steroid receptor has been demonstrated in women with endometriosis. It has been shown not only for estrogen receptors (ER), whose dominance is increased during the secretory phase [38,44], but also for progesterone receptors (PR), reducing decidualization capacity [14]. However, progesterone signaling via its receptor is an absolute requirement for implantation and pregnancy. Indeed, it activates the transcription of genes involved in ovulation, uterine receptivity, implantation, decidualization, and pregnancy maintenance. Progesterone downregulates steroid receptors, such as ER and PR, in the epithelial cells during the mid-secretory phase [26]. Such downregulation can be delayed when the hormone level or response is reduced, as observed in luteal phase defects [12]. 

The PR is expressed as PR-A and PR-B isoforms, which function differently. Progesterone action on target genes is conferred primarily by PR-B. PR-A is a dominant inhibitor of the transcriptional activity of PR-B. Progesterone resistance is primarily due to an imbalance between the two isoforms. Indeed, the presence of the inhibitory isoform PR-A and the absence of the stimulatory PR-B induce a significant reduction in PR in endometriotic implants compared with the normal endometrium. This situation can be explained by epigenetic mechanisms, where hypermethylation of the promoter region of PR-B inactivates it. Furthermore, it seems to be graded in response to the immunologic characteristics of the patient, as well as the amount, activity, and location of disease [13,25,31]. 

In the normal menstrual cycle, levels of the PR gradually increase in the endometrium during the follicular phase, peaking immediately before, and decreasing after ovulation, suggesting that estradiol stimulates PR levels. This process is essential for the transition of the endometrium from a proliferative to a secretory and receptive stage. At the time of endometrial receptivity, progesterone normally down-regulates both endometrial PR and ER, through the induction of 17β-hydroxysteroid dehydrogenase type 2 [14]. In contrast, in endometriosis, there is an inadequate response to progesterone of both the eutopic and ectopic endometrial cells and tissue. This is demonstrated by low expression of PR-B (contributing to increased local levels of estradiol), blunted expression of progesterone target genes, and an inadequate decidualization response [31,38]. Associated with inflammatory changes in the endometrium, progesterone resistance is categorized as the hallmark for implantation failure [14].

Nevertheless, studies have challenged the theory of defective implantation capacity as the probable cause of endometriosis-associated infertility [23]. In particular, Garcia-Velasco et al. used the endometrial receptivity array (ERA), which evaluates the expression of 238 specific genes directly related to endometrial receptivity during the WOI, in order to evaluate the endometrial receptivity of patients with different stages of endometriosis, as well as in healthy controls [45]. Endometrial samples were collected during laparoscopic surgery from infertile women with endometriosis (*n* = 17) and without endometriosis (*n* = 5). The stage of endometriosis was based on the revised staging system of the American Society for Reproductive Medicine [46]. Seven patients had minimal endometriosis (41%), three patients had mild endometriosis (17%), four patients presented moderate disease (23%) and three patients had severe disease (17%). Five classes of samples based on the endometriosis stage (I, II, III, IV, controls) were analyzed. None of the 238 genes present in the ERA array were significantly over or under expressed, thereby suggesting that endometrial receptivity is similar between women with and without endometriosis, as well as across different stages of endometriosis [23,45].

In conclusion, progesterone resistance results in inadequate antagonism of estrogen action, increased inflammation, inadequate stromal differentiation and endometrial remodeling that may lead to an unreceptive endometrium for embryo implantation [38]. Currently, studies have challenged the theory of defective implantation capacity and have suggested that endometrial receptivity is similar between women with and without endometriosis. Gene expression is only one factor. Other genes that may not been studied or epigenetic aberrations might provide further insight into the understanding of endometriosis [45].

## 7. Pregnancy Outcomes

Women with endometriosis seem to face greater obstetric risks than the general population [47]. Maternal complications may include increased risk of pre-eclampsia [48], antepartum [48]/postpartum hemorrhage [49,50], placenta praevia (PP) [49,51,52,53,54,55], and caesarean section [37,48,50,52], whereas fetal problems include prematurity [48,51,52,56], small for gestational age [52,53,56], or macrosomia [53], and neonatal intensive care unit admission [53]. The association between endometriosis and spontaneous miscarriages has been long debated, without reaching a consensus. Santulli et al. showed a significantly increased incidence rate ratio for miscarriages in women with endometriosis [57], confirmed by recent systematic reviews and meta-analysis [52,58].

Regarding PP, a retrospective cohort study [47] assessed pregnancy outcomes in 419 women who achieved a first spontaneous singleton pregnancy after surgery for endometriosis, and found a 3.7% incidence of PP in women with endometriosis, more than ten times the figure of 0.3% reported in the general population, especially in the case of rectovaginal lesions (7.6%, i.e., 25-fold increase) [37]. Takemura et al. confirmed the increased risk of PP in the case of endometriosis after ART [59]. A systematic review also observed that patients with endometriosis undergoing ART might have an additional risk of PP (three-fold higher after ART in endometriosis, compared to patients without endometriosis after ART) [60]. ART itself has been associated with a six-fold increased risk of PP [60]. The rate of PP was similar for FET and fresh ET, according to a systematic review and meta-analysis in 2018 [61]. On the contrary, a more recent systematic review and meta-analysis (2022) showed that FET is associated with a decreased rate of PP compared to fresh ET [62]. A retrospective study determined an association between endometrial thickness (more than 12 mm) and PP, compared to women with endometrial thickness less than 9 mm [55]. Previous studies reported that endometrial thickness was thinner in the case of FET than in the case of fresh ET [55,62]. Therefore, a possible mechanism for the decreased rate of PP is the thinner endometrial thickness in the case of FET [62]. 

Furthermore, PP is associated with an increased rate of placenta accreta spectrum disorder (PASD) and is the most significant risk factor for PASD. Therefore, it is possible that women with endometriosis have a higher prevalence of PASD than those without endometriosis. This was confirmed by a systematic review and meta-analysis, which assessed the relationship between PASD and endometriosis [62], especially in the case of ART and FET [61,62].

## 8. Associated Adenomyosis

Studies have shown a strong association between adenomyosis and endometriosis, reaching 54–90% of cases, especially in younger women, suggesting a common pathogenesis and demonstrating the close relationship between the two pathologies [63]. 

Adenomyosis is defined by the presence of endometrial glands and stroma within the myometrium, combined with reactional myometrial hypertrophia [64], causing an alteration in the endometrium–myometrium interface, the so-called Junctional Zone (JZ) [18,19]. Its etiology is currently unknown. The invasion of the myometrium by endometrial tissue through an altered or absent JZ and positive feedback via TIAR (tissue injuries and repair) mechanism was often proposed [65,66], as well as the *de novo* theory, involving endometrial tissue development from embryonic mullerian remnants or adult stem cells present into the myometrium. Migration of endometrial cells due to menstrual reflux is also considered in the case of deep endometriosis to explain the focal adenomyosis of the outer myometrium [18,67]. Adenomyosis can be present as diffuse, focal or cystic adenomyosis [68].

The clinical symptoms of adenomyosis are increased menstrual flow, dysmenorrhea, abnormal uterine bleeding and chronic pelvic pain. One third of the patients are asymptomatic [69]. Currently, it is thought to be a specific form of endometriosis and is defined as endometriosis of the uterus [19], although other authors identify adenomyosis and endometriosis as separate entities. 

While in the past, adenomyosis was diagnosed during a hysterectomy, recent progress in imaging allows detection in younger women. However, its diagnosis is still difficult and adenomyosis is underreported. Transvaginal ultrasound and pelvic magnetic resonance imaging are non-invasive diagnostic tools, but there is a lack of consensus on the criteria to use, even though MUSA criteria (Morphological Uterus Sonographic Assessment) tend to be widely followed [70,71]. Definitive diagnostics depend on surgery and pathology assessment, with the evaluation of myometrial tissue samples showing endometrial invasion, which is not feasible in patients wishing to conceive.

A relationship between adenomyosis and infertility has been postulated, since adenomyosis is found in approximately 25% of infertile women. Due to its location, adenomyosis could have an impact on the chances of pregnancy, especially for patients with implantation disorders, rather than oocyte quality. 

One mechanism is the abnormal contractility, a consequence of the alteration of the myometrial part of the junctional zone, called the archi-myometrium [19]. The hypertrophied muscle fibers observed in adenomyosis promote marked peristalsis and increased endometrial pressure, which seems to affect sperm transport and implantation [18,24].

Another mechanism could be the excessive chronic inflammation with altered vascularization and an improper secretion of cytokines. Furthermore, several enzymes that produce ROS are overexpressed, leading to abnormal levels of free radicals that could damage the early embryo, its implantation and development [18,28].

Studies have also demonstrated endometrial receptor changes in patients with adenomyosis. Indeed, ER expression is increased by the inflammation, and expression of PR is reduced in the junctional zone, inducing progesterone resistance. Another factor, P450 aromatase, is overexpressed, as in endometriosis, leading to increased local endometrial estrogen production, altering the balance between estrogens and progesterone [18,28]. 

For the process called decidualization, the expression of factors involved in the suppression of the immune response is necessary. In adenomyosis, their concentration does not increase in the secretory phase, as in healthy patients. Likewise, overexpression of ER results in the decreased production of adhesive molecules (such as integrins) and implantation markers (such as HOXA10 and LIF); however, these molecules must be at high levels for a successful interaction between the embryo and endometrium [18,19,72].

All aforementioned mechanisms could explain the association between adenomyosis and implantation failure. 

A meta-analysis published in 2014 supports the negative influence of adenomyosis on the implantation of good quality embryos in patients undergoing IVF treatment with autologous oocytes. The authors observed a 28% reduction in the likelihood of clinical pregnancy in women with infertility having adenomyosis, when compared to women without adenomyosis. Miscarriage rates are also increased in women with adenomyosis (regardless of maternal age and genetic status of embryos) [19,73]. 

In 2005, Piver showed that JZ thickness is a good predictor of implantation success. The greater the JZ thickness, the lower the pregnancy rate per transfer [74].

As adenomyosis is a primarily uterine disease and could be associated with implantation failure, this might, therefore, interfere with implantation in endometriosis patients [19,22]. In 2014, Vercellini et al., for example, describe the adverse impact of adenomyosis on fertility in deep endometriosis [75].

Nevertheless, further studies are required to determine the reason of implantation failure in women with adenomyosis and the impact of adenomyosis on infertile women with or without endometriosis.

## 9. Conclusions

Endometriosis is associated with a highly inflammatory, pro-angiogenic and hormone-rich microenvironment, which may impair embryo implantation by several possible mechanisms, which are as follows: oocyte alterations and poor embryo quality, hormonal imbalance, inadequate expression of various endometrial receptivity molecules, or progesterone resistance (Figure 1).

Controversial results in the literature add even more difficulties to the understanding of the mechanisms that lead to embryo implantation disorders. Depending on the research group, data are conflicting. Furthermore, either oocyte/embryo impairment, altered endometrium, or both may cause impaired implantation in patients with endometriosis. New prospective, randomized, and controlled studies are necessary to determine the authentic origin of the defects that make conception more difficult in the case of endometriosis and adenomyosis. Caution is needed in extrapolating the results of studies with the ART protocol to the general population. Indeed, a possible impact of the ovarian stimulation on the development of endometriosis or on the cumulative live birth rate between fresh and FET, a different environment of IVF and natural conception and the presence of a luteal support are multiple factors that can impact a study’s results. Better understanding of the different mechanisms and interactions involved in endometriosis (and adenomyosis) could help us to find effective therapies. The debate continues. 

## Figures and Tables

**Figure 1 jcm-11-05366-f001:**
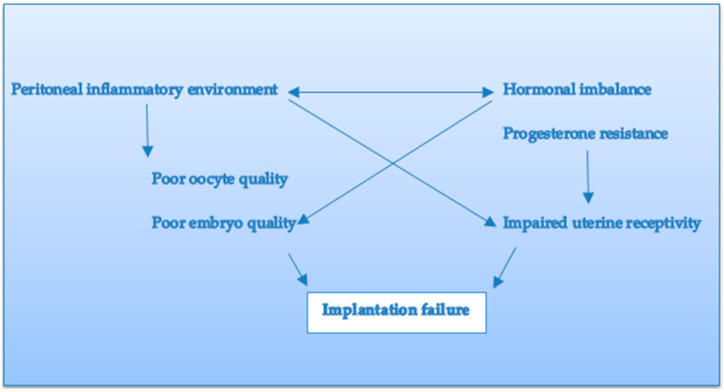
Possible mechanisms underlying embryo implantation in endometriosis.

## Data Availability

Not applicable.

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
