# Peer review of "Implantation Failure in Endometriosis Patients: Etiopathogenesis"

_jcm, 2022, doi:10.3390/jcm11185366_

Round 1

Reviewer 1 Report

Thank  you  for the opportunity to review this article.

In this manuscript, the authors present a narrative review of implantation failure in women with endometriosis. This is an important and interesting subject; however, in my view, the following concerns should be addressed before this manuscript can be considered for publication:

1. The abstract needs to be rewritten; results should be mentioned, and the authors’ opinions should be summarized in a discussion/conclusion section in the Abstract . The current Abstract does not summarize the results of the review and will not be useful to readers.

2. The authors have described the association between endometriosis and endometrial implantation. I recommend that in each section, it is necessary to add a discussion of the measures adopted to address the concerns highlighted in the study, the lack of ongoing clinical trials, which serves as a limitation, and future perspectives for each issue.

3. Line 63: I suggest “Embryo Implantation” should be used as MeSH keywords.

4. Line 67: Please clarify the studies identified.

Although this is an old study, authors should mention the following study, which has been cited more than 200 times in previous research and significantly contributes to the literature in this field: Fertil Steril. 1996 Mar;65(3):603-7.

5. I recommend that the authors discuss the association between endometrial thickness and endometriosis. A recent systematic review reported that endometriosis was associated with an increased incidence of placenta accreta spectrum. In my opinion,  endometriosis associated with low endometrial thickness at embryo implantation may lead to an increased incidence of placenta accreta spectrum.

6. Figure 1: I recommend that authors please recheck Figure 1 before submission. Please remove the underlined with a red wavy line.

Reviewer 2 Report

I have read with great interest the proposed manuscript. It represent a good summary of the recent literature concerning the selected topic. I still find it to  slightly misleading dur to other not mentioned studies that actually report the contrary. Regarding the oocyte quality - I think the authors should include those publications and also to trie to balance the conclusions. This is one example by Juneau et al 10.1016/j.fertnstert.2017.05.038.  Regarding the endometrial receptivity there are several good quality publications reporting equal outcomes when using frozen embryo transfer. I think that should also be included in the text and the text should be more balanced, as this is a common therapeutic proposition.

Round 2

Reviewer 1 Report

The authors have addressed all of my previous comments.